# DIFFUSION MODELS WITHOUT CLASSIFIER-FREE GUIDANCE

## ABSTRACT

We introduce Model-guidance (MG), a novel training objective for diffusion models that addresses the limitations of the widely used Classifier-free Guidance (CFG). Our approach directly incorporates the posterior probability of conditions into training, allowing the model itself to act as an implicit classifier. MG is conceptually inspired by CFG yet remains simple and effective, serving as a plug-and-play module compatible with existing architectures. Our method significantly accelerates training and doubles inference speed by requiring only a single forward pass per denoising step. MG achieves generation quality on par with, or surpassing, state-of-the-art CFG-based diffusion models. Extensive experiments across multiple models and datasets demonstrate both the efficiency and scalability of our approach. Notably, MG achieves a state-of-the-art FID of 1.34 on the ImageNet 256 benchmark.

## 1 INTRODUCTION

Diffusion models (Sohl-Dickstein et al., 2015; Song & Ermon, 2019; Ho et al., 2020; Song et al., 2021a;b) have become the cornerstone of many successful generative models, *e.g.* , image generation (Dhariwal & Nichol, 2021; Nichol et al., 2022; Rombach et al., 2022; Podell et al., 2024; Chen et al., 2024) and video generation (Ho et al., 2022; Blattmann et al., 2023; Gupta et al., 2025; Polyak et al., 2024; Wang et al., 2024) tasks. However, diffusion models also struggle to generate "low temperature" samples (Ho & Salimans, 2021; Karras et al., 2024a) due to the nature of training objectives, and techniques such as Classifier guidance (Dhariwal & Nichol, 2021) and Classifier-free Guidance (CFG) (Ho & Salimans, 2021) are proposed to improve performances.

Despite its advantage and ubiquity, CFG has several drawbacks (Karras et al., 2024a) and poses challenges to effective implementations (Kynkäänniemi et al., 2024) of diffusion models. One critical limitation is the simultaneous modeling of unconditional task apart from the conditional task during inference. The unconditional model is typically implemented by randomly dropping the condition of training pairs and replacing with an manually defined empty label. The introduction of additional tasks may reduce network capabilities and lead to skewed sampling distributions (Karras et al., 2024a; Kynkäänniemi et al., 2024). Furthermore, CFG requires two forward passes per denoising step during inference, one for the conditioned and another for the unconditioned model, thereby significantly escalating the computational costs.

In this work, we propose Model-guidance (MG), an innovative method for diffusion models to effectively circumvent CFG and boost performances, thereby eliminating the limitations above. We propose a novel objective that transcends from simply modeling the data distribution to incorporating the posterior probability of conditions. Specifically, we leverage the model itself as an implicit classifier and directly learn the score of calibrated distribution during training.

As depicted in fig. 1, our proposed method confers multiple substantial breakthroughs. It significantly refines generation quality and accelerates both training and inference processes, with experiments showcasing significant improvements over both vanilla and CFG diffusion models. Specifically, the inference speed is doubled with our method, as each denoising step needs only one network forward in contrast to two in CFG. Besides, it is easy to implement and requires only one line of code modification, making it a plug-and-play module of existing diffusion models with instant improvements. Finally, it is an end-to-end method that excels traditional two-stage distillation-based approaches and even outperforms other models that use CFG.

(a) $\nabla_{x_t} \log \tilde{p}_\theta(x_t|c) = \nabla_{x_t} \log p_\theta(x_t|c) p_\theta(c|x_t)^w \propto \nabla_{x_t} \log p_\theta(x_t|c) \left( \frac{p_\theta(x_t|c)}{p_\theta(x_t)} \right)^w$

(b) `target += w * (model(x_t, t, c) - model(x_t, t, zeros)).detach()`

Figure 1: **We propose Model-guidance (MG), remove Classifier-free Guidance (CFG) for diffusion models and achieve state-of-the-art on ImageNet 256 generation with FID of $1.34$.**
**(a)** While CFG needs two forwards (green and red), MG directly produces the final output (blue).
**(b)** MG requires few line of code modification while providing significant quality improvements.
**(c)** Comparing to concurrent methods, whether or not using CFG, MG yields the lowest FID results.
**(d)** In addition to superior quality, the sampling cost of MG is also much lower than other methods.

We conduct comprehensive experiments on the prevalent Imagenet (Deng et al., 2009; Russakovsky et al., 2015) benchmarks with $256 \times 256$ and $512 \times 512$ resolution, compare with a wide variates of concurrent methods, and scale up to text-to-image models to attest the effectiveness of our proposed method. The evaluation results demonstrate that our method not only parallels and even outperforms other approaches that uses CFG, but also scales to different models and datasets, making it a promising enhancement for diffusion models. In conclusion, we make the following contribution:

- We proposed a novel method, Model-guidance (MG), to effectively train diffusion models.
- MG removes CFG for diffusion models and greatly accelerates both training and inference.
- Extensive experiments with SOTA results on ImageNet demonstrate the advantages of MG.

## 2 BACKGROUND

### 2.1 DIFFUSION AND FLOW MODELS

**Diffusion models** (Sohl-Dickstein et al., 2015; Song & Ermon, 2019; Ho et al., 2020) are a class of generative models that utilize forward and reverse stochastic processes to model complex data distributions. The forward process adds noise and transforms data samples into Gaussian distributions

$$q(x_t|x_0) = \mathcal{N}\left(x_t; \sqrt{\bar{\alpha}_t}x_0, (1 - \bar{\alpha}_t)\mathbf{I}\right), \tag{1}$$

where $x_t$ represents the noised data at timestep $t$ and $\bar{\alpha}_t = \prod_{s=1}^{t} \alpha_s$ is the noise schedule. Conversely, the reverse process learns to denoise and finally recover the original data, which aims to reconstruct score (Sohl-Dickstein et al., 2015; Song et al., 2021b) from the noisy samples $x_t$ by

$$p_\theta(x_{t-1}|x_t) = \mathcal{N}\left(x_{t-1}; \mu_\theta(x_t, t), \Sigma_\theta(x_t, t)\right), \tag{2}$$

where $\mu_\theta$ and $\Sigma_\theta$ are mean and variance and predicted by neural networks. In common implementations, the training of diffusion models leverages a re-parameterized objective (Ho et al., 2020)

$$\mathcal{L}_{\text{simple}} = \mathbb{E}_{t,x_0,\epsilon} \|\epsilon_\theta(x_t, t) - \epsilon\|^2. \tag{3}$$

**Flow Models** (Lipman et al., 2023; Liu et al., 2023; Albergo et al., 2023; Tong et al., 2024), an emerging type of generative models, utilize the concept of Ordinary Differential Equations (ODEs) and a forward process of Optimal Transport (OT) interpolant (McCann, 1997). They learn the directions from noise pointing to ground-truth data by optimize

$$\mathcal{L}_{\text{FM}} = \mathbb{E}_{t,x_0,\epsilon} \|u_\theta(x_t) - u_t(x_t|x_0)\|^2, \quad \text{where } x_t = (1-t)x_0 + t\epsilon \text{ and } u_t(x_t|x_0) = x_0 - \epsilon. \tag{4}$$

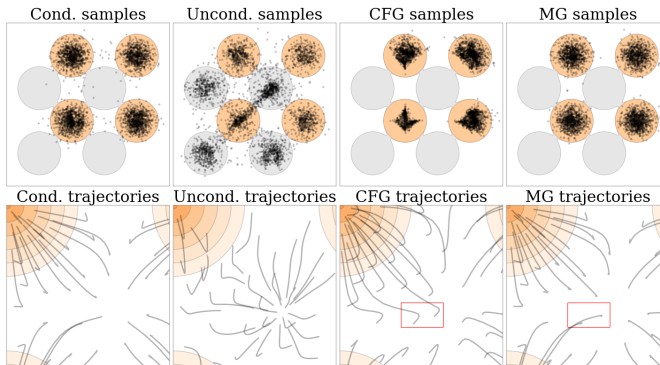

Figure 2: We use a grid 2D distribution with two classes, orange and gray, as an example and train diffusion models. **(a)** In the first row, CFG improves quality by eliminating the outliers, while the samples concentrate in the center and thus lost diversity. In contrast, MG yields less outliers, better coverage of data, and higher diversity than CFG. **(b)** In the second row, the trajectories of CFG turn sharply at the beginning, *e.g.* the samples inside the red box, while our method directly drives them to the closest data distributions.

## 2.2 CLASSIFIER-FREE GUIDANCE

Due to the complex nature of visual datasets, diffusion models often struggle whether to recover real image distribution or engage in the alignment to conditions. Classifier-free Guidance (CFG) (Ho & Salimans, 2021) is proposed and has become indispensable for modern diffusion models (Nichol & Dhariwal, 2021; Karras et al., 2022; Saharia et al., 2022; Hoogeboom et al., 2023).

The key design of CFG is to combine the posterior probability and utilize Bayes' rule during inference time. To facilitate this, it is required to train both conditional and unconditional diffusion models. In particular, CFG trains the models to predict

$$\epsilon_\theta(x_t, t, c) = -\sigma_t \nabla_{x_t} \log p_\theta(x_t|c) \quad \text{and} \quad \epsilon_\theta(x_t, t, \varnothing) = -\sigma_t \nabla_{x_t} \log p_\theta(x_t), \tag{5}$$

where an additional empty class $\varnothing$ introduced in common practices. During training, the model switches between the two modes with a ratio $\lambda$. For inference, the model combines the conditional and unconditional scores on-the-fly and guides the denoising process as

$$\tilde{\epsilon}_\theta(x_t, t, c) = \epsilon_\theta(x_t, t, c) + w \cdot (\epsilon_\theta(x_t, t, c) - \epsilon_\theta(x_t, t, \varnothing)), \tag{6}$$

where $w$ is the guidance scale that controls the focus on conditional scores and the trade-off between generation quality and sampling diversity.

However, CFG has with several disadvantages (Karras et al., 2024a; Kynkäänniemi et al., 2024), such as the multitask learning of both conditional and unconditional generation, and the doubled inference cost due to the doubled number of function evaluations (NFEs). Moreover, the tempting property that solving the denoising process according to eq. (6) eventually recovers data distribution does not hold, as the joint distribution does not represent a valid heat diffusion of the ground-truth (Zheng & Lan, 2024). This results in exaggerated truncation and mode dropping similar to (Karras et al., 2018; Brock et al., 2019; Sauer et al., 2022), since the samples are blindly pushed towards the regions with higher posterior probability. As fig. 2 shows, CFG improves sample quality as the cost of diversity lost and distort trajectories (Karras et al., 2024a).

## 2.3 DISTILLATION-BASED METHODS

Besides acceleration (Song et al., 2023; Liu et al., 2023), researchers also adopt distillation on diffusion models with CFG to improve sampling quality. GD (Meng et al., 2023) learns a smaller one-step model to match the performance of larger multi-step models, while ADD (Sauer et al., 2024) additionally introduces an adversarial approach. Pioneering diffusion models (Black-Forest-Labs, 2024; Stability-AI, 2024) are released with distilled versions. However, these approaches involve two-stage learning and require extra computation and storage for offline teacher models.

## 3 METHOD

### 3.1 MODEL-GUIDANCE LOSS

Instead of direct sampling from the learned conditional distribution, the core idea of CFG originates from the classifier-guidance (Dhariwal & Nichol, 2021) and samples from the joint distribution

$$\tilde{p}_\theta(x_t|c) \propto p_\theta(x_t|c)p_\theta(c|x_t)^w, \tag{7}$$

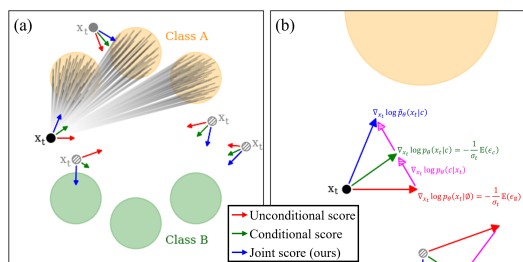

Figure 3: Illustration of CFG and proposed MG. **(a)** The conditional and unconditional scores point towards the centroids of data distribution, as the training pairs $(x_0, \epsilon)$ are randomly sampled. **(b)** While CFG provides better update directions by interpolating the two vectors during inference, MG aims to directly learn the joint score, $\nabla \log \tilde{p}_\theta(x_t|c)$, during training and thereby reduces inference costs.

where $w$ is the weighting factor of posterior probability. Since the classifier $p_\theta(c|x_t)$ of noisy samples is not practically available, CFG propose to estimate with Bayes' rule on-the-fly in inference

$$p_\theta(c|x_t) = \frac{p_\theta(x_t|c)p_\theta(c)}{p_\theta(x_t)}, \tag{8}$$

where $p_\theta(x_t|c)$ and $p_\theta(x_t)$ are conditional and unconditional distributions. However, we argue that do we really need to separately learn $p_\theta(x_t|c)$ and $p_\theta(x_t)$? Can we fuse the joint distribution and turn the diffusion model itself as an implicit classifier in a more *efficient* and *elegant* way?

As fig. 3 illustrates, we propose Model-guidance (MG) to directly learn the joint distribution $\tilde{p}_\theta(x_t|c)$, of which the score is decomposed as

$$\underbrace{\nabla_{x_t} \log \tilde{p}_\theta(x_t|c)}_{\text{joint score}} = \underbrace{\nabla_{x_t} \log p_\theta(x_t|c)}_{\text{conditional score}} + w \cdot \underbrace{\nabla_{x_t} \log p_\theta(c|x_t)}_{\text{posterior score}} \tag{9}$$

During training, the ground-truth of the conditional score (Song & Ermon, 2019) in eq. (9) is

$$\nabla_{x_t} \log p_\theta(x_t|c) = -\frac{1}{\sigma_t}\epsilon, \tag{10}$$

where $\epsilon$ is the noise added to samples. However, the ground-truth of the posterior score, *e.g.* the score of posterior probability $p_\theta(c|x_t)$, cannot be directly obtained. Inspired by eq. (8), we transform the diffusion model itself into an implicit classifier. Specifically, we employ Bayes' rule during training to estimate the posterior probability

$$\log p_\theta(c|x_t) = \log p_\theta(x_t|c) - \log p_\theta(x_t) + \log p_\theta(c)$$
$$\propto \log p_\theta(x_t|c) - \log p_\theta(x_t) \tag{11}$$

Next, we use the diffusion model itself to approximate the scores

$$\nabla_{x_t} \log p_t(x_t|c) = -\frac{1}{\sigma_t}\epsilon_\theta(x_t, t, c) \quad \text{and} \quad \nabla_{x_t} \log p_t(x_t) = -\frac{1}{\sigma_t}\epsilon_\theta(x_t, t, \varnothing), \tag{12}$$

where $\sigma_t$ is the variance of the noise added to $x_t$ at timestep $t$, $\varnothing$ is the empty class, and $\epsilon_\theta(\cdot)$ is the diffusion model. Substituting eq. (12) into eq. (11) yields

$$\nabla_{x_t} \log p_\theta(c|x_t) \propto -\frac{1}{\sigma_t}\left(\epsilon_\theta(x_t, t, c) - \epsilon_\theta(x_t, t, \varnothing)\right). \tag{13}$$

Combining eqs. (10) and (13) leads to a valid training target for eq. (9)

$$\nabla_{x_t} \log \tilde{p}_\theta(x_t|c) = -\frac{1}{\sigma_t}(\epsilon + w\left(\epsilon_\theta(x_t, t, c) - \epsilon_\theta(x_t, t, \varnothing)\right)). \tag{14}$$

Then, our method applies the Bayes' estimation in eq. (11) online and trains a conditional diffusion model to directly predict the score in eq. (9) according to eq. (14), instead of separately learning eq. (12) in the form of vanilla CFG. Consequently, our MG loss is

$$\mathcal{L}_{\text{MG}} = \mathbb{E}_{t,(x_0,c),\epsilon}\|\epsilon_\theta(x_t, t, c) - \epsilon'\|^2, \tag{15}$$
$$\epsilon' = \epsilon + w \cdot \text{sg}(\epsilon_\theta(x_t, t, c) - \epsilon_\theta(x_t, t, \varnothing)), \tag{16}$$

where we apply the stop gradient operation, $\text{sg}(\cdot)$, as a common practice of avoiding model collapse (Grill et al., 2020). We come across with the surprising observation that our MG loss behaves similar to form of self-supervise learning (Grill et al., 2020) that the diffusion model $\epsilon_\theta$ provides a calibrated prediction target in eq. (16), while itself is also being trained with the same target in

---

**Algorithm 1** Training with Model-guidance loss

**Input:** Dataset $\{\mathbf{X_i}, \mathbf{C_i}\}$, noise schedule $\bar{\alpha}$, learning rate $\eta$
**Output:** Model $\epsilon_\theta$
**repeat**
    Sample data $(x_0, c) \sim \{\mathbf{X_i}, \mathbf{C_i}\}$
    Sample noise $\epsilon \sim \mathcal{N}(0, 1)$ and time $t \sim \mathbf{U}(0, 1)$
    Add noise with $x_t = \sqrt{\bar{\alpha}_t} x_0 + \sqrt{1 - \bar{\alpha}_t} \epsilon$
    Modify target $\epsilon' = \epsilon + w \cdot \mathrm{sg}(\epsilon_\theta(x_t, c, t) - \epsilon_\theta(x_t, \varnothing, t))$
    Compute loss $\mathcal{L}_{\mathrm{MG}} = \|\epsilon_\theta(x_t, c, t) - \epsilon'\|^2$
    Back propagation $\theta = \theta - \eta \nabla_\theta \mathcal{L}_{\mathrm{MG}}$
**until** converged

---

eq. (15), producing a cycle of self improvement. Therefore, MG accelerates the training and convergence of diffusion models, comparing to the vanilla forms in eq. (3). For flow-based models, we also have the similar objective

$$\mathcal{L}_{\mathrm{MG}} = \mathbb{E}_{t,(x_0,c),\epsilon} \|u_\theta(x_t, t, c) - u'\|^2, \tag{17}$$

$$u' = u + w \cdot \mathrm{sg}(u_\theta(x_t, t, c) - u_\theta(x_t, t, \varnothing)). \tag{18}$$

where $u$ is the ground-truth flow in eq. (4). During training, we randomly drop and replace the condition $c$ in eqs. (15) and (17) with an additional empty class $\varnothing$ by a certain ratio $\lambda$ to enable the estimation in eq. (12), similar to the common implementations of CFG. MG transforms the model itself into an online classifier during training, jointly optimizes generation quality and condition alignment, and is compatible with existing pipelines with minimum modifications.

## 3.2 COMPARE MG WITH DISTILLATION

The questions arise that what is the essential difference between MG and distillation? Specifically, we compare MG with guidance distillation (GD) (Meng et al., 2023), and investigate the advantages of MG in depth. Given a pre-trained diffusion model $\epsilon_\theta(x_t, t, c)$ as teacher, GD introduces another student model $\epsilon_\eta(x_t, t, c, w)$, which takes the guidance scale $w$ as input, and optimize with

$$\mathcal{L}_{\mathrm{GD}} = \mathbb{E}_{w,t,(x_0,c),\epsilon} \|\epsilon_\eta(x_t, t, c, w) - \epsilon_\theta^w(x_t, t, c)\|^2, \tag{19}$$

$$\epsilon_\theta^w(x_t, t, c) = \epsilon_\theta(x_t, t, c) + w(\epsilon_\theta(x_t, t, c) - \epsilon_\theta(x_t, t, \varnothing)), \tag{20}$$

where $w \sim \mathbf{U}[w_{\min}, w_{\max}]$ is sampled from a range of guidance scales we are interested in. With $\mathcal{L}_{\mathrm{GD}}$, the teacher model $\epsilon_\theta(x_t, t, c)$ is offline and fixed during the training process, and only the student model $\epsilon_\eta(x_t, t, c, w)$ benefits from CFG. The distilled model $\epsilon_\eta$ is upperbounded and cannot surpass its teacher $\epsilon_\theta$. In contrast, our MG uses the model itself as an online teacher in eqs. (15) and (16). The model benefits from the guidance and continues to produce better guidance for itself.

## 3.3 IMPLEMENTATION VARIANTS

**Scale-aware networks.** Similar to other distillation-based methods (Frans et al., 2024; Meng et al., 2023), the guidance scale $w$ can be fed into the network as an additional condition. In particular, we sample guidance scale from an specified interval, and the loss objective is modified as the following

$$\mathcal{L}_{\mathrm{MG}} = \mathbb{E}_{w,t,(x_0,c),\epsilon} \|\epsilon_\theta(x_t, t, c, w) - \epsilon'\|^2, \tag{21}$$

$$\epsilon' = \epsilon + w \cdot \mathrm{sg}(\epsilon_\theta(x_t, t, c, 0) - \epsilon_\theta(x_t, t, \varnothing, -1)). \tag{22}$$

When augmented with $w$-input, our models offer flexible choices of the balance between image quality and sample diversity during inference, and still require only one forward per denoising step.

**Automatic adjustment of the hyper-parameter $w$.** While the scale $w$ in eqs. (16) and (18) plays an important role as the balance of posterior probability, it is tedious and costly to perform manual search during training. Therefore, we introduce an automatic adjustment scheme of $w$. We begin with $w = 0$ that corresponds to vanilla diffusion models in eq. (3), then update $w$ according to intermediate evaluations. The value is raised when quality decreases and suppressed otherwise, leading to an optimums when converged. We also provide the details and pseudo-code in Appedix.

Table 1: Experiments on ImageNet 256 conditional generation without CFG. By employing MG, the performances of both DiT and SiT are greatly boosted, achieving state-of-the-art.

| Model | FID↓ | sFID↓ | IS↑ | Pre.↑ | Rec.↑ | Tflops↓ | #Params↓ |
|---|---|---|---|---|---|---|---|
| ADM (Dhariwal & Nichol, 2021) | 10.9 | - | 101.0 | 0.69 | 0.63 | 1120 | 554M |
| VDM++ (Kingma & Gao, 2023) | 2.40 | - | 225.3 | 0.78 | 0.66 | - | 2.0B |
| LDM (Rombach et al., 2022) | 10.5 | - | 103.5 | 0.71 | 0.62 | 103.6 | 400M |
| U-ViT (Bao et al., 2023) | 8.97 | - | 136.7 | 0.69 | 0.63 | - | 501M |
| MDTv2 (Gao et al., 2023) | 5.06 | - | 155.6 | 0.72 | 0.66 | - | 676M |
| REPA (Yu et al., 2024b) | 5.90 | 6.33 | 162.1 | 0.71 | 0.56 | 29.65 | 675M |
| LightningDiT (Yao & Wang, 2025) | 2.17 | 4.36 | 205.6 | 0.77 | 0.65 | 131.2 | 675M |
| VAR (Tian et al., 2024) | 2.16 | - | 288.7 | 0.81 | 0.61 | - | 2.0B |
| RAR (Yu et al., 2024a) | 3.83 | - | 274.5 | 0.79 | 0.61 | - | 1.5B |
| MAR (Li et al., 2024) | 2.35 | - | 227.8 | 0.79 | 0.62 | - | 943M |
| DiT-XL/2 (Peebles & Xie, 2023) | 9.62 | 6.85 | 121.5 | 0.67 | 0.67 | 29.65 | 675M |
| +MG(ours) | **1.78** | 4.46 | 298.5 | 0.80 | 0.66 | 29.65 | 675M |
| Improvement | 81.5% | 34.9% | 146% | 19.4% | −0.1% | 0.0% | 0.0% |
| SiT-XL/2 (Ma et al., 2024) | 8.61 | 6.32 | 131.7 | 0.68 | 0.67 | 29.65 | 675M |
| +MG(ours) | **1.34** | 4.58 | 321.5 | 0.81 | 0.65 | 29.65 | 675M |
| Improvement | 84.4% | 27.5% | 144% | −19.1% | −3.0% | 0.0% | 0.0% |

Table 2: Experiments on ImageNet 256 conditional generation with CFG. Comparing to models using CFG, our method still obtains excellent results and surpasses others with lower Tflops.

| Model | FID↓ | sFID↓ | IS↑ | Pre.↑ | Rec.↑ | Tflops↓ | #Params↓ |
|---|---|---|---|---|---|---|---|
| ADM (Dhariwal & Nichol, 2021) | 4.59 | 5.25 | 186.7 | 0.82 | 0.52 | 2240 | 544M |
| VDM++ (Kingma & Gao, 2023) | 2.12 | - | 267.7 | 0.81 | 0.65 | - | 2.0B |
| LDM (Rombach et al., 2022) | 3.60 | - | 247.7 | 0.87 | 0.48 | 207.2 | 400M |
| U-ViT (Bao et al., 2023) | 2.29 | 5.68 | 263.9 | 0.82 | 0.57 | - | 501M |
| MDTv2 (Gao et al., 2023) | 1.58 | 4.52 | 314.7 | 0.79 | 0.65 | - | 676M |
| REPA (Yu et al., 2024b) | 1.42 | 4.70 | 305.7 | 0.80 | 0.65 | 59.30 | 675M |
| LightningDiT (Yao & Wang, 2025) | 1.35 | 4.15 | 295.3 | 0.79 | 0.65 | 262.3 | 675M |
| VAR (Tian et al., 2024) | 1.73 | - | 350.2 | 0.82 | 0.60 | - | 2.0B |
| RAR (Yu et al., 2024a) | 1.48 | - | 326.0 | 0.80 | 0.63 | - | 1.5B |
| MAR (Li et al., 2024) | 1.55 | - | 303.7 | 0.81 | 0.62 | - | 943M |
| DiT-XL/2 (Peebles & Xie, 2023) | 2.27 | 4.60 | 278.2 | 0.83 | 0.57 | 59.30 | 675M |
| +MG(ours) | **1.78** | 4.46 | 298.5 | 0.80 | 0.66 | 29.65 | 675M |
| Improvement | 21.6% | 3.04% | 7.30% | −3.6% | 15.8% | 100% | 0.0% |
| SiT-XL/2 (Ma et al., 2024) | 2.06 | 4.49 | 277.5 | 0.83 | 0.59 | 59.30 | 675M |
| +MG(ours) | **1.34** | 4.58 | 321.5 | 0.81 | 0.65 | 29.65 | 675M |
| Improvement | 35.0% | −2.0% | 15.9% | −2.4% | 10.2% | 100% | 0.0% |

## 4 EXPERIMENT

We first present a system-level comparison with state-of-the-art models on ImageNet $256 \times 256$ conditional generation. Then we conduct ablation experiments to investigate the detained designs of our method. Especially, we emphasize on the following questions:

- How far can MG push the generation quality of existing diffusion models? (tables 1 and 2, section 4.2)
- Can MG scales to larger models, more datasets, and different image generation tasks? (tables 3 to 5, section 4.3)
- What is the details about implementation, flexibility, and efficiency of MG? (tables 6 to 8, figs. 4 and 5, section 4.4)

### 4.1 SETUP

**Implementation and dataset.** We follow the experiment pipelines in DiT (Peebles & Xie, 2023) and SiT (Ma et al., 2024). We use ImageNet (Deng et al., 2009; Russakovsky et al., 2015) dataset and the Stable Diffusion (Rombach et al., 2022) VAE to encode $256 \times 256$ images into the latent space of $\mathbb{R}^{32 \times 32 \times 4}$. We conduct ablation experiments with the B/2 variant of DiT and SiT models

Table 3: Experiments on model size.

| Model | FID↓ | sFID↓ | IS↑ | Pre.↑ | Rec.↑ |
|---|---|---|---|---|---|
| DiT-B/2 | 43.5 | 36.7 | 39.23 | 0.62 | 0.34 |
| +CFG | 9.67 | 9.14 | 160.6 | 0.79 | 0.36 |
| +MG$_{(ours)}$ | **7.24** | 5.56 | 189.2 | 0.84 | 0.38 |
| DiT-L/2 | 23.3 | 18.4 | 132.7 | 0.73 | 0.40 |
| +CFG | 6.76 | 7.10 | 203.4 | 0.78 | 0.38 |
| +MG$_{(ours)}$ | **5.43** | 4.66 | 236.3 | 0.83 | 0.44 |
| DiT-XL/2 | 19.5 | 15.6 | 163.5 | 0.79 | 0.46 |
| +CFG | 4.16 | 5.23 | 233.0 | 0.82 | 0.49 |
| +MG$_{(ours)}$ | **3.37** | 4.73 | 257.2 | 0.84 | 0.51 |
| SiT-B/2 | 33.0 | 27.8 | 65.24 | 0.68 | 0.35 |
| +CFG | 8.35 | 8.63 | 173.0 | 0.78 | 0.37 |
| +MG$_{(ours)}$ | **6.49** | 5.69 | 212.3 | 0.86 | 0.38 |
| SiT-L/2 | 18.9 | 16.3 | 173.2 | 0.71 | 0.42 |
| +CFG | 5.77 | 6.46 | 223.8 | 0.79 | 0.44 |
| +MG$_{(ours)}$ | **4.50** | 4.03 | 243.9 | 0.85 | 0.46 |
| SiT-XL/2 | 17.3 | 13.9 | 192.1 | 0.78 | 0.50 |
| +CFG | 3.47 | 4.21 | 247.9 | 0.83 | 0.53 |
| +MG$_{(ours)}$ | **2.89** | 3.12 | 261.0 | 0.85 | 0.54 |

Table 4: Experiments on ImageNet 512.

| Model | FID↓ | sFID↓ | IS↑ | Pre.↑ | Rec.↑ |
|---|---|---|---|---|---|
| DiT-XL/2 | 12.0 | 7.12 | 105.3 | 0.75 | 0.64 |
| +CFG | 3.04 | 5.02 | 240.8 | 0.84 | 0.54 |
| +MG$_{(ours)}$ | **2.78** | 4.86 | 257.2 | 0.83 | 0.58 |
| SiT-XL/2 | 9.64 | 6.03 | 124.4 | 0.77 | 0.65 |
| +CFG | 2.62 | 4.18 | 252.2 | 0.84 | 0.57 |
| +MG$_{(ours)}$ | **2.24** | 4.03 | 276.9 | 0.86 | 0.60 |
| EDM2$_{XXL}$ | 1.91 | 4.65 | 226.7 | 0.80 | 0.68 |
| +CFG | 1.81 | 4.24 | 273.2 | 0.85 | 0.63 |
| +MG$_{(ours)}$ | **1.43** | 4.03 | 299.1 | 0.84 | 0.64 |

Table 5: Experiments on text-to-image models.

| Model | FID↓ (w/o CFG) | FID↓ (w/ CFG) |
|---|---|---|
| GLIDE (Nichol et al., 2022) | - | 12.32 |
| LDM (Rombach et al., 2022) | - | 12.58 |
| DALL·E 2 (Ramesh et al., 2022) | - | 10.46 |
| SD 1.5 (Rombach et al., 2022) | 23.11 | 8.32 |
| +MG$_{(ours)}$ | **7.86** | - |

Table 6: Experiments on scale $w$.

| Model | $w$ | FID↓ | sFID↓ | IS↑ | Pre.↑ | Rec.↑ |
|---|---|---|---|---|---|---|
| DiT-B/2 | 0.00 | 43.5 | 36.7 | 39.23 | 0.62 | 0.34 |
| +CFG | 2.00 | 9.67 | 9.14 | 160.6 | 0.79 | 0.36 |
| +MG$_{(ours)}$ | 0.25 | 9.86 | 8.87 | 176.1 | 0.81 | 0.37 |
| +MG$_{(ours)}$ | 0.50 | **7.24** | 5.56 | 189.2 | 0.84 | 0.38 |
| +MG$_{(ours)}$ | 0.75 | 8.21 | 6.63 | 197.2 | 0.86 | 0.38 |
| +MG$_{(ours)}$ | 1.00 | 9.66 | 7.90 | 224.7 | 0.85 | 0.39 |
| +MG$_{(ours)}$ | Auto | 7.60 | 6.29 | 192.4 | 0.85 | 0.38 |
| SiT-B/2 | 0.00 | 33.0 | 27.8 | 65.24 | 0.68 | 0.35 |
| +CFG | 2.00 | 8.35 | 8.63 | 173.0 | 0.78 | 0.37 |
| +MG$_{(ours)}$ | 0.25 | 8.94 | 7.87 | 194.3 | 0.83 | 0.38 |
| +MG$_{(ours)}$ | 0.50 | **6.49** | 5.69 | 212.3 | 0.86 | 0.38 |
| +MG$_{(ours)}$ | 0.75 | 8.03 | 6.91 | 221.0 | 0.86 | 0.39 |
| +MG$_{(ours)}$ | 1.00 | 9.14 | 7.99 | 236.7 | 0.88 | 0.40 |
| +MG$_{(ours)}$ | Auto | 6.86 | 5.88 | 219.1 | 0.87 | 0.38 |

Table 7: Experiments on scale-aware networks.

| Model | $w$-in | FID↓ | sFID↓ | IS↑ | Pre.↑ | Rec.↑ |
|---|---|---|---|---|---|---|
| DiT-B/2 | ✗ | 43.5 | 36.7 | 39.23 | 0.62 | 0.34 |
| +CFG | ✗ | 9.67 | 9.14 | 160.6 | 0.79 | 0.36 |
| +MG$_{(ours)}$ | ✗ | **7.24** | 5.56 | 189.2 | 0.84 | 0.38 |
| +MG$_{(ours)}$ | ✓ | 8.13 | 6.03 | 175.1 | 0.84 | 0.39 |
| SiT-B/2 | ✗ | 33.0 | 27.8 | 65.24 | 0.68 | 0.35 |
| +CFG | ✗ | 8.35 | 8.63 | 173.0 | 0.78 | 0.37 |
| +MG$_{(ours)}$ | ✗ | **6.49** | 5.69 | 212.3 | 0.86 | 0.38 |
| +MG$_{(ours)}$ | ✓ | 7.33 | 5.96 | 207.4 | 0.85 | 0.38 |

Table 8: Experiments on drop ratio $\lambda$.

| Model | $\lambda$ | FID↓ | sFID↓ | IS↑ | Pre.↑ | Rec.↑ |
|---|---|---|---|---|---|---|
| DiT-B/2 | 0.10 | 43.5 | 36.7 | 39.23 | 0.62 | 0.34 |
| +CFG | 0.10 | 9.67 | 9.14 | 160.6 | 0.79 | 0.36 |
| +MG$_{(ours)}$ | 0.05 | 11.7 | 9.90 | 156.7 | 0.78 | 0.33 |
| +MG$_{(ours)}$ | 0.10 | **7.24** | 5.56 | 189.2 | 0.84 | 0.38 |
| +MG$_{(ours)}$ | 0.15 | 7.62 | 5.99 | 183.4 | 0.83 | 0.38 |
| +MG$_{(ours)}$ | 0.20 | 9.01 | 7.04 | 171.7 | 0.81 | 0.36 |
| SiT-B/2 | 0.10 | 33.0 | 27.8 | 65.24 | 0.68 | 0.35 |
| +CFG | 0.10 | 8.35 | 8.63 | 173.0 | 0.78 | 0.37 |
| +MG$_{(ours)}$ | 0.05 | 10.8 | 9.25 | 168.8 | 0.80 | 0.34 |
| +MG$_{(ours)}$ | 0.10 | **6.49** | 5.69 | 212.3 | 0.86 | 0.38 |
| +MG$_{(ours)}$ | 0.15 | 6.77 | 5.89 | 207.4 | 0.85 | 0.37 |
| +MG$_{(ours)}$ | 0.20 | 8.87 | 8.06 | 199.6 | 0.84 | 0.37 |

and train for 400K iterations. During training, we use AdamW (Kingma, 2014; Loshchilov, 2019) optimizer and a batch size of 256 in consistent with DiT (Peebles & Xie, 2023) and SiT (Ma et al., 2024) for fair comparisons. For inference, we use 250 sampling steps for both DiT and SiT models and Euler-Maruyama sampler for SiT.

**Baseline Models.** We compare with several state-of-the-art image generation models, including both diffusion-based and AR-based methods, which can be classified into the following three classes: (a) *Pixel-space diffusion*: ADM (Dhariwal & Nichol, 2021), VDM++ (Kingma & Gao, 2023); (b) *Latent-space diffusion*: LDM (Rombach et al., 2022), U-ViT (Bao et al., 2023), MDTv2 (Gao et al., 2023), REPA (Yu et al., 2024b), LightningDiT (Yao & Wang, 2025), DiT (Peebles & Xie, 2023), SiT (Ma et al., 2024); (c) *Auto-regressive models*: VAR (Tian et al., 2024), RAR (Yu et al., 2024a), MAR (Li et al., 2024). These models consist of strong baselines for experiment evaluations.

**Evaluation metrics.** We report the commonly used Frechet inception distance (FID) (Heusel et al., 2017). In addition, we report sFID (Nash et al., 2021), Inception Score (IS) (Salimans et al., 2016), Precision (Pre.), and Recall (Rec.) (Kynkäänniemi et al., 2019) as supplementary metrics. We also report the computation costs to generate one sample with each model in terms of FLOPS to measure the trade-off between generation quality and inference speed.

## 4.2 OVERALL EVALUATION

**System-level performance.** We present a thorough comparison with recent state-of-the-art image generation approaches on ImageNet $256 \times 256$ dataset in tables 1 and 2. As shown in table 1, both

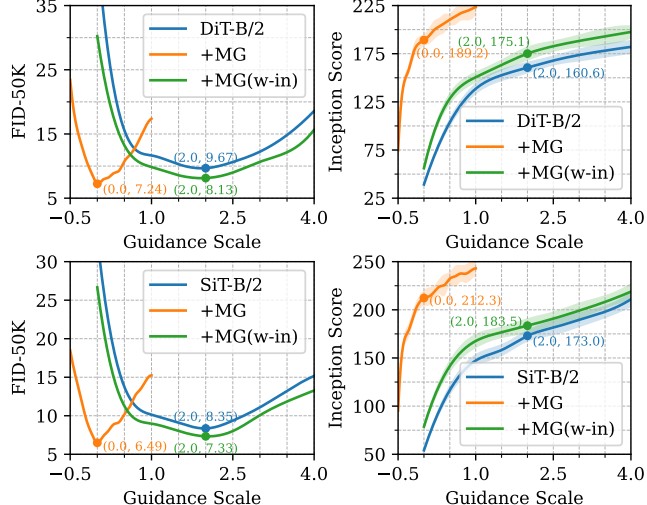

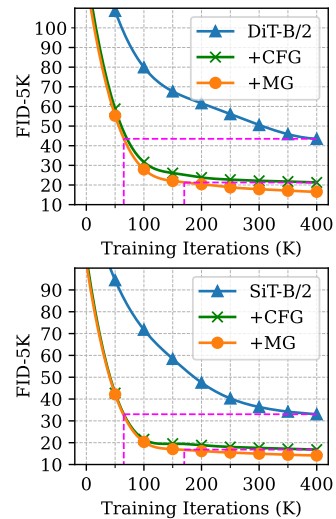

Figure 4: FID-50K and Inception Score with different scales during inference. MG also offers inference-time flexibility to control the tradeoff of quality and diversity.

Figure 5: FID-5K results during training. MG is faster than both vanilla and CFG models.

DiT-XL/2 and SiT-XL/2 models greatly benefit from our method, achieving the outstanding performance gain of $81.5\%$ and $84.4\%$. It is worth mentioning that our models do not apply modern techniques in the inference process, including rejection sampling (Tian et al., 2024), Classifier-free Guidance (Ho et al., 2020) and guidance interval (Kynkäänniemi et al., 2024). Compared to advanced methods, our models are light-weight, *e.g.* 675M in contrast to RAR-XXL with 1.5B and MAR-H with 943M parameters, and consume less computational resources, for example, LightningDiT uses DiT-XL/1 to reduce patch size to $1 \times 1$ and needs $16\times$ computation in attention operations. We visualize our results in fig. 6.

**Compare with CFG.** To facilitate a fair evaluation, we also compare with other methods that uses Classifier-free Guidance. While other methods significantly benefit and are indispensable from CFG, it introduces an additional forward without condition and doubles the sampling costs. Also, it usually requires a careful search over the hyper-parameter of guidance scale to achieve the best trade-off between quality and diversity. In contrast, our models still surpass other methods with CFG.

**Sampling cost.** We report the computation consumption for each model to generate one sample in terms of Tflops. Other methods uses Classifier-free Guidance to boost quality and require to run models twice during inference, one with condition and one without condition. Meanwhile, our models run significantly faster and do not sacrifice inference speed for sampling quality.

## 4.3 SCALING UP

**Model sizes.** We study MG with different sizes of models in table 3. We train the B/2, L/2, and XL/2 variants of DiT and SiT models, and compare with both vanilla and CFG results. It demonstrates that our method is capable to boost the performance of models with different sizes and designs.

**High-resolution images.** Next, we apply MG on ImageNet $512 \times 512$ dataset in table 4 to validate our method in handling high resolution images and difficult distributions. In addition to DiT and SiT, we also conduct experiments with EDM2-XXL (Karras et al., 2024b). The results confirm that the generation quality is also significantly improved with MG and exceeds CFG by a remarkable margin in all cases.

**Text-to-image models.** Furthermore, scaling up our method to text-to-image models is of imparable significance. In table 5, we finetune Stable Diffusion (SD) 1.5 (Rombach et al., 2022) model with MG on LAION-400M (Schuhmann et al., 2021) dataset, and report FID of zero-shot text-to-image generation on COCO 2014 (Lin et al., 2014) dataset. As shown, MG improves the FID of SD 1.5 from 23.11 to 7.86 and even surpasses the 8.32 of CFG. It reveals the promising capability of MG across various conditioning modalities as shown in fig. 7.

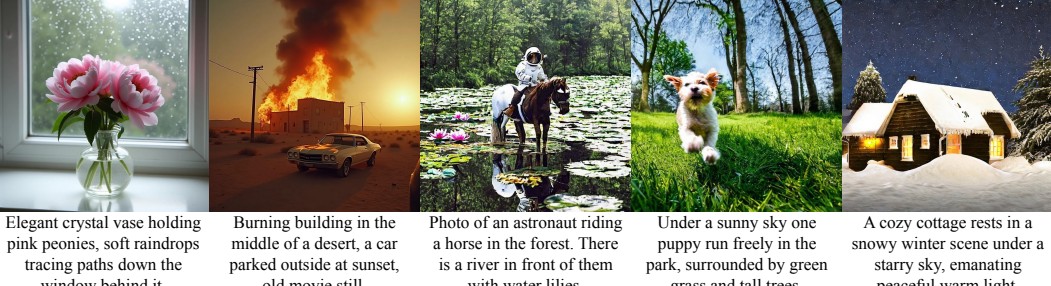

Figure 6: Uncurated samples of SiT-XL/2+MG.

| | | | | |
|---|---|---|---|---|
| Elegant crystal vase holding pink peonies, soft raindrops tracing paths down the window behind it. | Burning building in the middle of a desert, a car parked outside at sunset, old movie still. | Photo of an astronaut riding a horse in the forest. There is a river in front of them with water lilies. | Under a sunny sky one puppy run freely in the park, surrounded by green grass and tall trees. | A cozy cottage rests in a snowy winter scene under a starry sky, emanating peaceful warm light. |

Figure 7: Text-to-image samples of SD1.5+MG.

### 4.4 ABLATION STUDY

**Hyper-parameter** $w$**.** In eqs. (16) and (18), the hyper-parameter $w$ controls the scale of posterior probability and serves an important role akin to the guidance scale in CFG. We conduct ablation experiments with different choices of $w$ in table 6, where $w = 0$ refers to vanilla diffusion models. It shows that $w$ also acts as a crucial role and balances the trade-off between quality and diversity. Nevertheless, MG is still better than CFG even if $w$ is not optimized, *e.g.* $w = 0.75$.

**Auto-tuning** $w$**.** To overcome the tedious manual search of $w$, we propose to automatically adjust $w$, and obtain comparable performances with manual search in the highlighted rows in table 6.

**Scale-aware networks.** It is optional for MG whether to take the scale $w$ as an additional input. In table 7, the models with $w$-input (orange) slightly lag behind the counterparts without $w$-input (blue), but still exceeds the quality of CFG, demonstrating the superiority of our method.

**Hyper-parameter** $\lambda$**.** The dropping ratio $\lambda$ is important to our method. To enable the estimation in eq. (12), we randomly drop the condition $c$ of part of data, and replace it with an additional empty label $\varnothing$. In table 8, we find that $\lambda \in \{0.10, 0.15\}$ offers satisfactory performances.

**Flexibility.** In fig. 4, MG is also capable of sampling with different balances between quality and diversity. For MG with our proposed scale-aware networks, we can simply feed different guidance scales $w$ as input. Otherwise, we can also wrap MG models with standard CFG, exactly as vanilla diffusion models. More details are provided in Appendix.

**Efficiency.** One key advantage of MG is that it not only improves inference speed, but also accelerates the training process and convergence. In fig. 5, MG obtains significantly faster training speed and better performance comparing to both vanilla and CFG models. In terms of sampling efficiency, MG requires less FLOPS than other methods in tables 1 and 2 with performances in parallel with LightningDiT (Yao & Wang, 2025), which consumes $\approx 9\times$ computation resources than MG.

## 5 CONCLUSION

This work addresses the limitations of the commonly used Classifier-free Guidance (CFG) of diffusion models, and proposes Model-guidance (MG) as an efficient and advantageous replacement. We first investigate the mechanism of CFG and locate the source of performance gain as a joint optimization of posterior probability. Then, we transcend the idea into the training process of diffusion models and directly learn the score of the joint distribution, $\nabla \log \tilde{p}_\theta(x_t|c) = \nabla \log p_\theta(x_t|c)p_\theta(c|x_t)^w$. Comprehensive experiments demonstrate that our method significantly boosts the generation performance without efficiency loss, scales to different models and datasets, and achieves state-of-the-art results on ImageNet $256 \times 256$ dataset. We believe this work contributes to future diffusion models.

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

## A REPRODUCTION CODE

We provide the code to reproduce the SiT-XL/2+MG results with FID of $1.34$ in tables 1 and 2. Please kindly refer to our anonymous supplementary materials for further information.

## B LIMITATIONS AND BROADER IMPACTS

**Limitations** While this work aims to improve the generation quality and sampling speed of current diffusion models, it is still relied on existing encoders and decoders (Rombach et al., 2022). Since concurrent works (Yao & Wang, 2025) that propose to train encoders and decoders in an end-to-end manner have achieved great success, a natural and encouraging next step is to take both encoders and decoders into accounts.

**Broader Impacts** While this work is mainly conducted on image generation diffusion models, autoregressive (AR) models and generative models on other modalities also employ CFG to improve performance. Extending this work to AR models and other modalities is also promising. On the negative side, our method learns from its training data and could therefore mirror any biases present. The image generation capabilities in this work might also be misused to spread false information, and we will limit the release of our model weights.

## C COMPARE MG WITH DISTILLATION

### C.1 OFFLINE DISTILLATION

Table 9: Comparison between MG and Guided-Distillation.

| Model | FID↓ | sFID↓ | IS↑ | Pre.↑ | Rec.↑ |
|---|---|---|---|---|---|
| DiT-XL/2 | 9.62 | 6.85 | 121.5 | 0.67 | 0.67 |
| +CFG | 2.27 | 4.60 | 278.2 | 0.83 | 0.57 |
| +GD | 2.31 | 4.59 | 282.3 | 0.82 | 0.59 |
| +MG$_{(ours)}$ | **1.78** | 4.46 | 298.5 | 0.80 | 0.66 |
| SiT-XL/2 | 8.61 | 6.32 | 131.7 | 0.68 | 0.67 |
| +CFG | 2.06 | 4.49 | 277.5 | 0.83 | 0.59 |
| +GD | 2.09 | 4.67 | 275.4 | 0.81 | 0.56 |
| +MG$_{(ours)}$ | **1.34** | 4.58 | 321.5 | 0.81 | 0.65 |

To thoroughly investigate the advantages of our method, we also conduct experiments in section C.1 to compare with the commonly used two-stage Guided-Distillation (GD) (Meng et al., 2023). As shown, the performances of GD are slightly worse than the original models, while MG exhibits significant improvements over the original models. In addition, the training costs of GD are higher than MG since its two-stage nature.

### C.2 ONLINE DISTILLATION

Although GD is designed as an offline method to distill pre-trained models, the idea of incorporating the form of GD in eqs. (23) and (24) into the pre-training stage as online distillation naturally emerges.

$$\mathcal{L}_{\mathrm{GD}} = \mathbb{E}_{w,t,(x_0,c),\epsilon} \left\| \epsilon_\eta(x_t,t,c,w) - \epsilon_\theta^w(x_t,t,c) \right\|^2, \tag{23}$$

$$\epsilon_\theta^w(x_t,t,c) = \epsilon_\theta(x_t,t,c) + w(\epsilon_\theta(x_t,t,c) - \epsilon_\theta(x_t,t,\varnothing)), \tag{24}$$

However, the student model $\epsilon_\eta(x_t,t,c,w)$ additionally consumes $w$ as input, which is different from the teacher model $\epsilon_\theta(x_t,t,c)$. Here, we thoroughly discuss both options.

**Case 1**: We abandon the additional input $w$ and train $\epsilon_\theta(x_t, t, c)$. In this case, the training loss is

$$\mathcal{L}_1 = \mathbb{E}_{w,t,(x_0,c),\epsilon} \left[ \underbrace{\|\epsilon_\theta(x_t, t, c) - \epsilon\|^2}_{\text{first term}} \right.$$

$$\left. + \underbrace{\|\epsilon_\theta(x_t, t, c) - \text{sg}(\epsilon_\theta(x_t, t, c) + w(\epsilon_\theta(x_t, t, c) - \epsilon_\theta(x_t, t, \varnothing)))\|^2}_{\text{second term}} \right]. \tag{25}$$

With $\mathcal{L}_1$, the model is confused by multiple update directions. The first term in eq. (25) directs the model to predict the ground-truth noise $\epsilon$, while the second term in eq. (25) instructs the model to match the prediction modified by CFG. On the other hand, MG in eqs. (15) and (16) has the unified prediction target and update direction, leading to faster convergence and better performance.

**Case 2**: We adopt the form $\epsilon_\theta(x_t, t, c, w)$ and feed with the scale $w$. In this case, the training loss is

$$\mathcal{L}_2 = \mathbb{E}_{w,t,(x_0,c),\epsilon} \left[ \underbrace{\|\epsilon_\theta(x_t, t, c, 0) - \epsilon\|^2}_{\text{first term}} \right.$$

$$\left. + \underbrace{\|\epsilon_\theta(x_t, t, c, w) - \text{sg}(\epsilon_\theta(x_t, t, c, 0) + w(\epsilon_\theta(x_t, t, c, 0) - \epsilon_\theta(x_t, t, \varnothing, -1)))\|^2}_{\text{second term}} \right]. \tag{26}$$

With $\mathcal{L}_2$, the model is pretrained in the first term in eq. (26) and distilled in the second term in eq. (26) separately. The distilled model $\epsilon_\theta(x_t, t, c, w)$ is supervised by another teacher model $\epsilon_\theta(x_t, t, c, 0)$, resulting the same issue as two-staged approaches.

Table 10: Comparison between MG and online distillation.

| Model | FID↓ | sFID↓ | IS↑ | Pre.↑ | Rec.↑ |
|---|---|---|---|---|---|
| DiT-B/2 | 43.5 | 36.7 | 39.23 | 0.62 | 0.34 |
| +CFG | 9.67 | 9.14 | 160.6 | 0.79 | 0.36 |
| +$\mathcal{L}_1$ | 11.3 | 12.4 | 147.9 | 0.74 | 0.34 |
| +$\mathcal{L}_2$ | 10.1 | 10.9 | 157.1 | 0.78 | 0.36 |
| +MG$_{(ours)}$ | **7.24** | 5.56 | 189.2 | 0.84 | 0.38 |
| SiT-B/2 | 33.0 | 27.8 | 65.24 | 0.68 | 0.35 |
| +CFG | 8.35 | 8.63 | 173.0 | 0.78 | 0.37 |
| +$\mathcal{L}_1$ | 9.76 | 10.7 | 154.3 | 0.72 | 0.35 |
| +$\mathcal{L}_2$ | 8.38 | 8.94 | 169.6 | 0.76 | 0.36 |
| +MG$_{(ours)}$ | **6.49** | 5.69 | 212.3 | 0.86 | 0.38 |

**Experimental comparison**. We additionally conduct experiments with the loss functions in eqs. (25) and (26) and compare with MG in section C.2. As depicted, eq. (25) obtains the worst results, while MG outperforms both eqs. (25) and (26).

## D HYPERPARAMETER AND IMPLEMENTATION DETAILS

**Implementations.** We implement our method based on the code of DiT (Peebles & Xie, 2023) and SiT (Ma et al., 2024; Yu et al., 2024b). Throughout the experiments, we use the exact same structure as DiT (Peebles & Xie, 2023) and SiT (Ma et al., 2024), and add an embedding for scale-aware networks that sums to the condition and timestep embedding. We use AdamW (Kingma, 2014; Loshchilov, 2019) with constant learning rate of 1e-4, $(\beta_1, \beta_2) = (0.9, 0.999)$ without weight decay. We also pre-compute and save the latent vectors of images and use these latent vectors for training. As consequences, we only apply simple random horizontal flip as data augmentation. We use `stabilityai/sd-vae-ft-ema` for encoding and decoding images. The detailed hyper-parameter setup are provided in table 11.

Table 11: Setup for tables 1 and 2.

|  | DiT-XL/2 | SiT-XL/2 |
| --- | --- | --- |
| **Architecture** | | |
| Input dim. | $32 \times 32 \times 4$ | $32 \times 32 \times 4$ |
| Num. layers | 28 | 28 |
| Hidden dim. | 1,152 | 1,152 |
| Num. heads | 16 | 16 |
| **Hyperparameters** | | |
| $w$ | 1.3 | 1.45 |
| $\lambda$ | 0.1 | 0.1 |
| Scale-aware | ✗ | ✗ |
| **Optimization** | | |
| Prediction | $\epsilon$ | v |
| Batch size | 256 | 256 |
| Optimizer | AdamW | AdamW |
| lr | 0.0001 | 0.0001 |
| $(\beta_1, \beta_2)$ | (0.9, 0.999) | (0.9, 0.999) |
| **Inference** | | |
| Sampler | ADM (Dhariwal & Nichol, 2021) | Euler-Maruyama |
| Steps | 250 | 250 |

**Algorithm 2** Training with Model-guidance Loss: PyTorch-like Pseudo-code

```
def train(scale=1.0, beta_1=0.05, beta_2=0.05, auto_adjust_scale=False):
    for step, (x, c) in enumerate(dataloader):
        # sample random noise and timestep
        noise = torch.randn(x.shape)
        timestep = torch.randint(0, diffusion.num_timesteps, x.size(0))

        # sample x_t from x
        x_t = diffusion.q_sample(x, timestep, noise)

        # randomly drop labels
        z[torch.perm(len(z))[:len(z) // 10]] = 0

        # predict noise from x_t
        noise_pred = net(x_t, timestep, z)

        # compute learning target
        with torch.no_grad():
            pred_w_cond = noise_pred.detach()
            # one additional forward
            pred_wo_cond = net(x_t, timestep, torch.zeros_like(z))
            target = noise + (scale - 1) * (pred_w_cond - pred_wo_cond)

        # compute loss and optimize
        loss = ((noise_pred - target) ** 2).mean()
        loss.backward()
        opt.step()
        opt.zero_grad()

        # to automatically adjust scale
        if auto_adjust_scale and step % freq_eval == 0:
            fid_eval = evaluate(net)
            # raise scale if fid is lower than history record
            if fid_eval < history_fid:
                scale *= (1 + beta_1)
            else:
                scale *= (1 - beta_1)
            history_fid = history_fid * (1 - beta_2) + fid_eval * beta_2
```

**Sampler.** For DiT, we use the ADM (Dhariwal & Nichol, 2021) sampler with 250 steps the same as the original DiT paper (Peebles & Xie, 2023). For SiT, we use the Euler-Maruyama sampler with 250 steps the same as the original SiT paper (Ma et al., 2024).

**Computing resources.** We use 16x NVIDIA A100 40GB GPUs for experiments. We use a batch size of 256 and remain unchanged for all experiments.

**Pseudo-code.** We provide a torch-like pseudo-code of training models with MG in algorithm 2.

## E  TRAINING WALL-TIME

Table 12: Training speed-up comparison on DiT-B/2.

|  | MG *vs.* vanilla DiT | MG *vs.* DiT+CFG |
|---|---|---|
| Training speedup (iterations to same FID) | 6.50× | 2.07× |
| Training speedup (wall-time to same FID) | 5.71× | 1.77× |
| Performance gain (same iterations) | 62% | 34% |
| Performance gain (same wall time) | 57% | 24% |

Table 13: Training speed-up comparison on SiT-B/2.

|  | MG *vs.* vanilla SiT | MG *vs.* SiT+CFG |
|---|---|---|
| Training speedup (iterations to same FID) | 7.04× | 2.11× |
| Training speedup (wall-time to same FID) | 6.18× | 1.94× |
| Performance gain (same iterations) | 57% | 23% |
| Performance gain (same wall time) | 53% | 20% |

In addition to fig. 5, we also report the training speed-up in terms of both iterations and wall-time in tables 12 and 13. When comparing with vanilla diffusion models, MG achieves $\geq 5.7\times$ training speed-up and $\geq 50\%$ performance gains. When comparing diffusion models using CFG sampling, MG still yields $\geq 1.7\times$ training speed-up and $\geq 20\%$ performance gains, whether in terms of training iterations or wall-time.

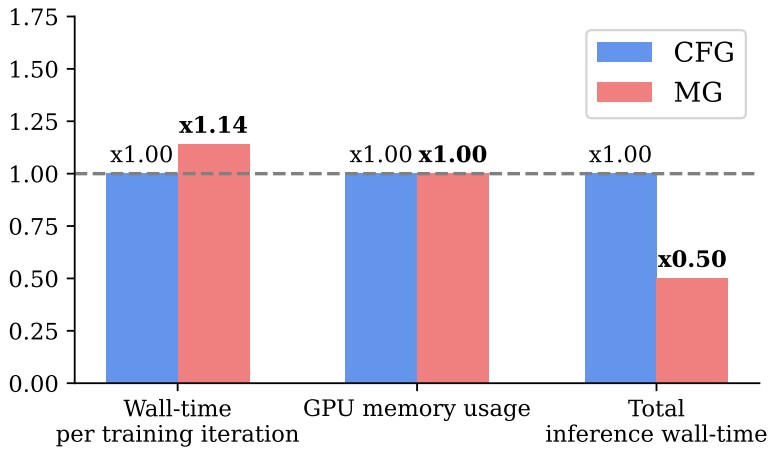

Figure 8: Comparing the computational efficiency of CFG and MG.

We also compare the wall-time per training iteration, GPU memory usage during training, and the total inference time to generate one image in fig. 8. Since MG requires only one additional forward of the online model during training (see algorithm 1), it significantly boosts convergence speed, improves performance, and reduce inference time by half at the cost of a mere overhead of $0.14\times$.

Note that while modern implementations of CFG concatenate conditional and unconditional predictions in the batch dimension and run one network forward with doubled batch size during inference,

Table 14: Comparison of total inference time in seconds to generate one sample. *CFG (separate)* means separately predicting conditional and unconditional scores during inference, and *CFG (parallel)* refers to concatenate conditional and unconditional predictions in batch dimension and forward once.

| Batch size | vanilla SiT | SiT+CFG (separate) | SiT+CFG (parallel) | SiT+MG |
|---|---|---|---|---|
| 32 | 1.42 | 2.86 | 2.86 | **1.42** |
| 64 | 1.37 | 2.76 | 2.75 | **1.37** |
| 128 | 1.33 | 2.67 | 2.64 | **1.33** |
| 256 | 1.30 | 2.61 | Out of Memory | **1.30** |

their inference time is still proportional to Tflops in tables 1 and 2 and thereby doubled than MG. In table 14, we also test and compare different implementations of CFG to MG on SiT-B/2 and one A100 80GB GPU with varying batch sizes. MG is twice faster than CFG even with parallel implementation and sufficient GPU memory.

Finally, we report the computational overhead of the proposed automatic adjustment algorithm for the hyper-parameter $w$. As in algorithm 2, we evaluate FID-1K as an intermediate evaluation metric to adjust scale, which introduces an inference process of 1 minutes per evaluation on DiT-B/2 and SiT-B/2 models. We evaluate and adjust every 20K iterations, which corresponds to an overhead of 20 minutes and $\approx 1.4\%$ of total training time.

## F INFERENCE-TIME FLEXIBILITY

### F.1 SCALE-AWARE NETWORKS

MG is also capable of sampling with different balances between quality and diversity. When implemented with the proposed scale-aware networks, we can simply change the input scale $w$. MG obtains better quality and diversity than CFG, as shown by the lower FID and higher IS scores across all guidance scale than CFG in fig. 4.

### F.2 APPLYING CFG ON MG MODELS

While MG itself achieves better results in most cases, MG still offers inference-time flexibility even without the proposed scale-aware networks by applying the standard CFG. Specifically, MG models $\epsilon_\theta$ are fully compatible with the original CFG framework. We can query the network twice to get $\epsilon_\theta(x_t, t, c)$ with condition and $\epsilon_\theta(x_t, t, \varnothing)$ with empty label, then compute the final prediction with $\tilde{\epsilon}_\theta(x_t, t, c) = \epsilon_\theta(x_t, t, c) + w(\epsilon_\theta(x_t, t, c) - \epsilon_\theta(x_t, t, \varnothing))$. Here, $w < 0$ is also a valid option.

### F.3 UNBIASED PREDICTION

The compatibility of MG and CFG leads to an important property. We can reformulate the MG loss in eqs. (15) and (16) as

$$
\begin{aligned}
\mathcal{L}_{\mathrm{MG}} &= \mathbb{E}_{t,(x_0,c),\epsilon} \left\| \epsilon_\theta(x_t, t, c) - [\epsilon + w \cdot \mathrm{sg}(\epsilon_\theta(x_t, t, c) - \epsilon_\theta(x_t, t, \varnothing))] \right\|^2 \\
&= \mathbb{E}_{t,(x_0,c),\epsilon} \left\| [\epsilon_\theta(x_t, t, c) - w \cdot \mathrm{sg}(\epsilon_\theta(x_t, t, c) - \epsilon_\theta(x_t, t, \varnothing))] - \epsilon \right\|^2,
\end{aligned}
\tag{27}
$$

which means that $[(1-w)\epsilon_\theta(x_t, t, c) + w\epsilon_\theta(x_t, t, \varnothing)]$ as a whole is trained by the standard diffusion loss in eq. (3) and learns the true distribution, *i.e.* $[(1-w)\epsilon_\theta^*(x_t, t, c) + w\epsilon_\theta^*(x_t, t, \varnothing)] = -\sigma_t \nabla_{x_t} \log p_t(x_t|c)$ when optimal. Then we can apply CFG with $w' = -w$ at inference time to get

$$
\begin{aligned}
\tilde{\epsilon}_{\theta^*}(x_t, t, c) &= \epsilon_{\theta^*}(x_t, t, c) + w'(\epsilon_{\theta^*}(x_t, t, c) - \epsilon_{\theta^*}(x_t, t, \varnothing)) \\
&= (1-w)\epsilon_{\theta^*}(x_t, t, c) + w\epsilon_{\theta^*}(x_t, t, \varnothing) \\
&= -\sigma_t \nabla_{x_t} \log p_t(x_t|c),
\end{aligned}
\tag{28}
$$

and sample with the unbiased prediction $\tilde{\epsilon}_{\theta^*}(x_t, t, c)$ to recover the true data distribution $p(x|c)$. This indicates that while MG drives the model to directly learn the joint distribution $\tilde{p}_\theta(x_t|c)$ in

eq. (7), the true data distribution $p(x|c)$ is implicitly learned and MG does not introduce diversity loss.

## F.4 COMPATIBILITY WITH OTHER METHODS

Table 15: Compatibility of MG and CADS (Sadat et al., 2024). CADS is also applicable on MG models.

|  | FID↓ |
| --- | --- |
| DiT (Peebles & Xie, 2023) | 9.62 |
| DiT+CFG (Peebles & Xie, 2023) | 2.27 |
| DiT+CADS (Sadat et al., 2024) | 1.70 |
| DiT+MG+CADS | **1.57** |

Besides the original CFG, MG is also orthogonal to and compatible with other techniques that improve DiT and SiT. As an example, we conduct ablation experiments in table 15 and show that MG is also orthogonal to and compatible with CADS (Sadat et al., 2024) and CADS also enjoys benefits from MG.

## G MORE RESULTS

### G.1 ADDITIONAL METRICS

Table 16: FD$_{\text{DINOv2}}$, KID, and CLIP-T evaluation results.

|  | FD$_{\text{DINOv2}}$↓ | KID↓ | CLIP-T↑ |
| --- | --- | --- | --- |
| REPA (Yu et al., 2024b) | 58.71 | 0.052 | 0.334 |
| LightningDiT (Yao & Wang, 2025) | 55.04 | 0.050 | 0.325 |
| DiT+CFG (Peebles & Xie, 2023) | 78.93 | 0.064 | 0.317 |
| SiT+CFG (Ma et al., 2024) | 70.52 | 0.053 | 0.326 |
| DiT+MG (ours) | 71.66 | 0.051 | 0.323 |
| SiT+MG (ours) | **53.49** | **0.043** | **0.342** |

Table 17: CLIP-T and HPSv2 evaluation results.

|  | CLIP-T↑ | HPSv2↑ |
| --- | --- | --- |
| Stable Diffusion (Rombach et al., 2022) 1.5 | 0.346 | 27.03 |
| Stable Diffusion 1.5 +MG (ours) | **0.352** | **27.18** |

In addition to the commonly used FID and IS, we also report more evaluation metrics and results, including FD$_{\text{DINOv2}}$ (Stein et al., 2023) in and Kernel Inception Distance (KID) (Bińkowski et al., 2018) in table 16. To assess the alignment of the generated images of MG and given conditions, we report CLIP-T (Radford et al., 2021) score in tables 16 and 17 by computing the CLIP similarity between images and the corresponding text prompts. The text prompt is "a photo of [CLASS]" for class-conditional image generation task on ImageNet in table 16, where [CLASS] is replaced by the label name. For text-to-image models, we additionally report the HPSv2 (Wu et al., 2023) results in table 17.

### G.2 2D EXAMPLES

We use a custom version of the visualization script in (Karras et al., 2024a) to train models and plot figs. 2 and 9 to 12. Beside the 2D example in fig. 2, we also provide more detailed examples in figs. 9, 11 and 12. The default scale of CFG is $w = 2$ in figs. 2, 9, 11 and 12, and we also provide illustrations with different scales in fig. 10.

## G.3 GENERATED SAMPLES

We present more visualization of generated samples with our SiT-XL/2+MG model in figs. 13 to 20. We also present more text-to-image samples of our SD1.5+MG model in fig. 21.

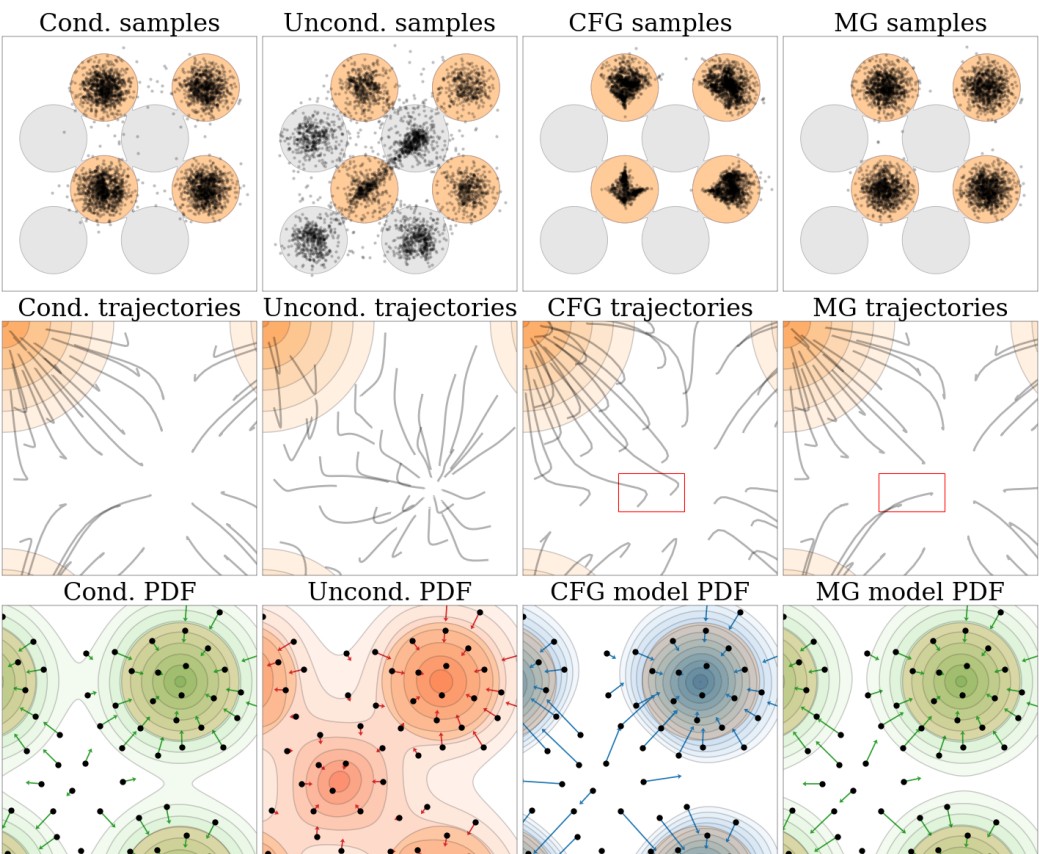

Figure 9: Diffusion models trained on a grid 2D distribution with two classes, which are marked with orange and gray colors, respectively. We plot the generated samples, trajectories, and probability density function (PDF) of conditional, unconditional, classifier-free guided model, and our approach.

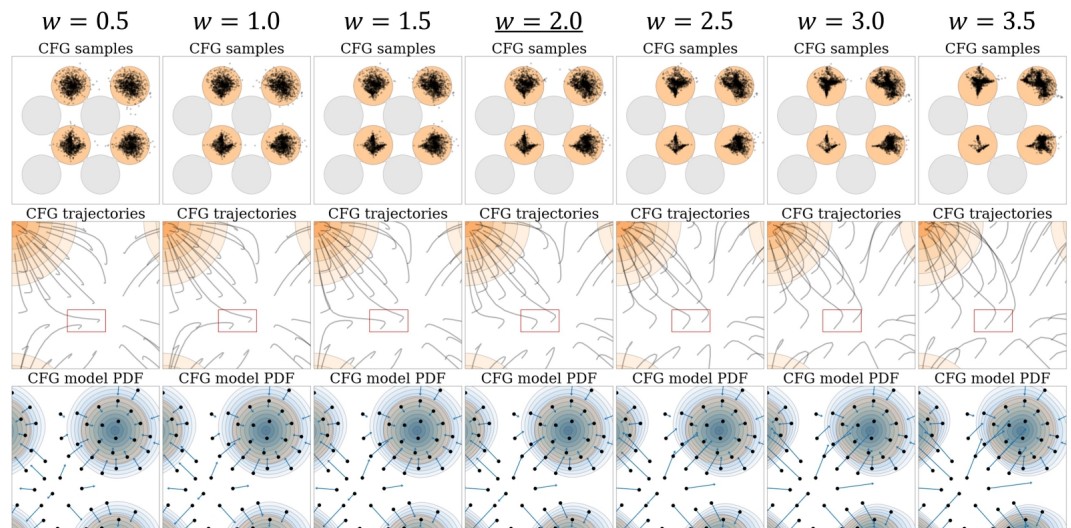

Figure 10: The generated samples, trajectories, and PDF at different guidance scales, where $w = 2$ corresponds to the CFG results in figs. 2 and 9.

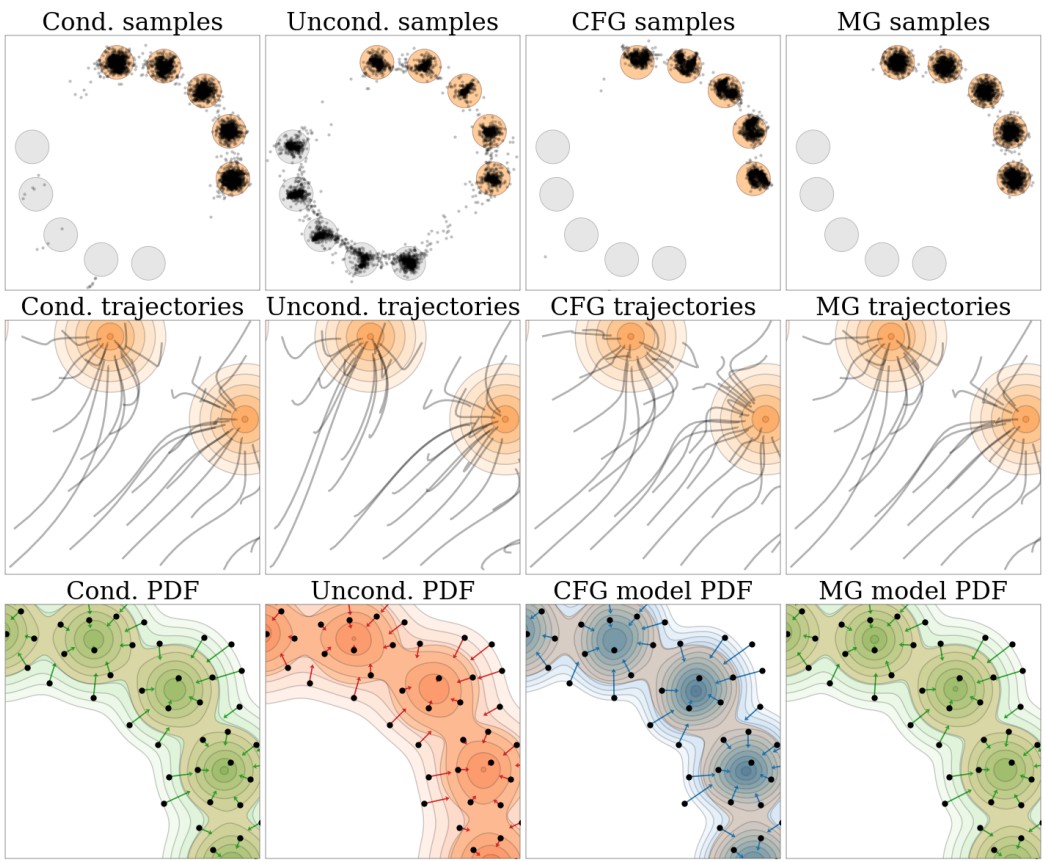

Figure 11: Diffusion models trained on a dots 2D distribution with two classes, which are marked with orange and gray colors, respectively. We plot the generated samples, trajectories, and probability density function (PDF) of conditional, unconditional, classifier-free guided model, and our approach.

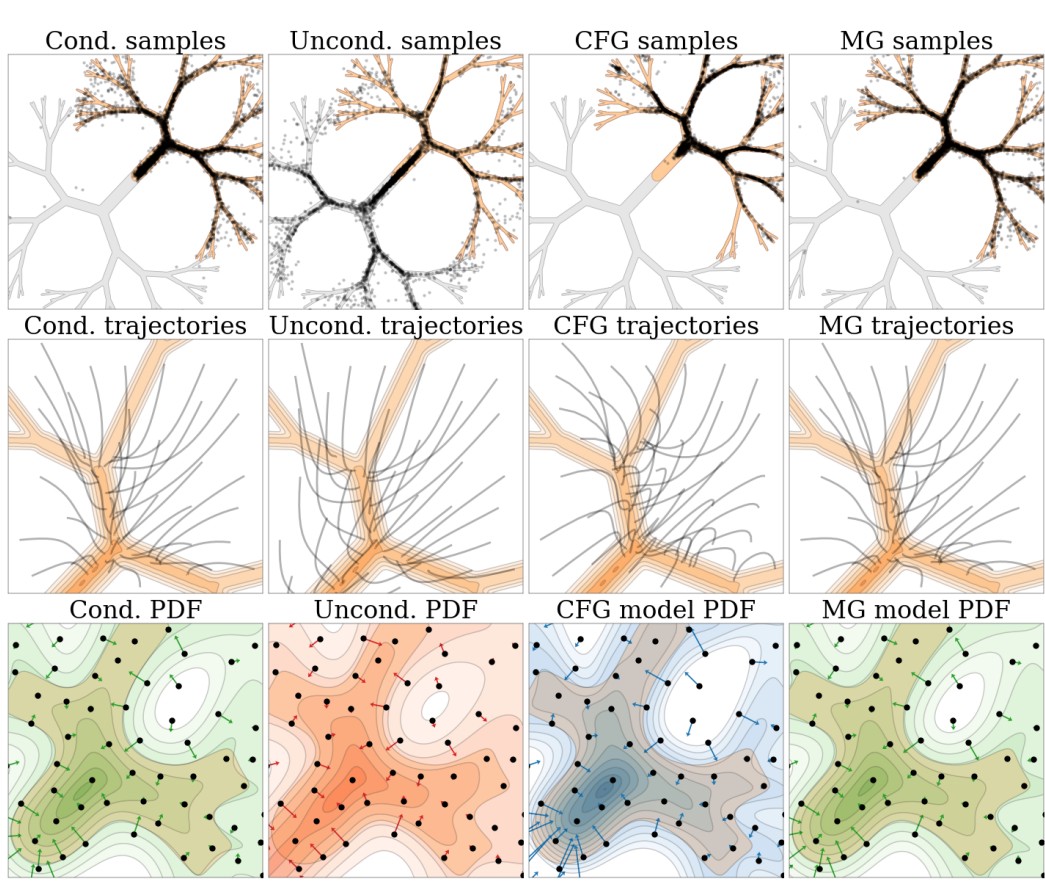

Figure 12: Diffusion models trained on a tree 2D distribution with two classes, which are marked with orange and gray colors, respectively. We plot the generated samples, trajectories, and probability density function (PDF) of conditional, unconditional, classifier-free guided model, and our approach.

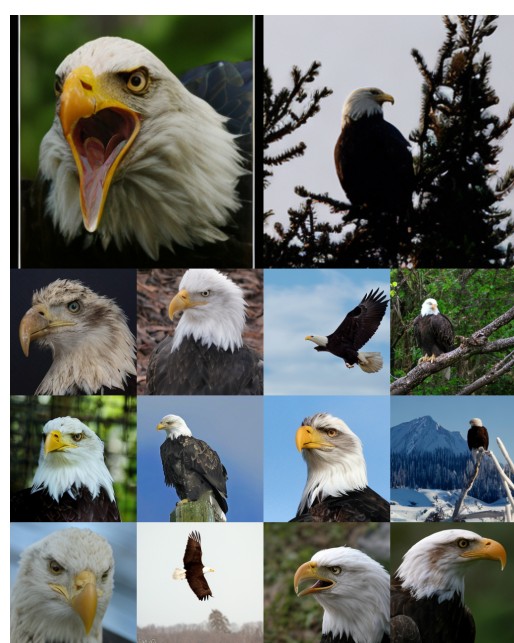

Figure 13: Generated samples of SiT-XL/2+MG on the class American eagle (22).

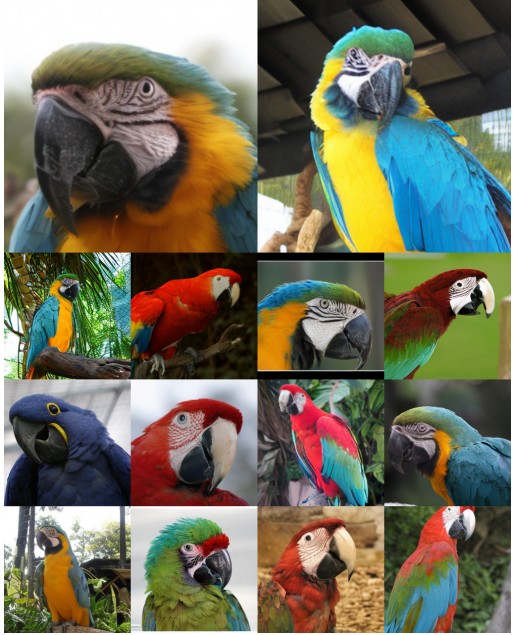

Figure 14: Generated samples of SiT-XL/2+MG on the class macaw (88).

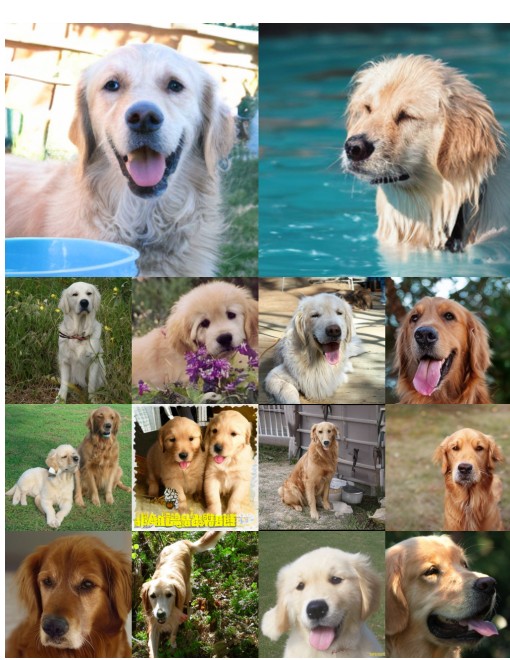

Figure 15: Generated samples of SiT-XL/2+MG on the class golden retriever (207).

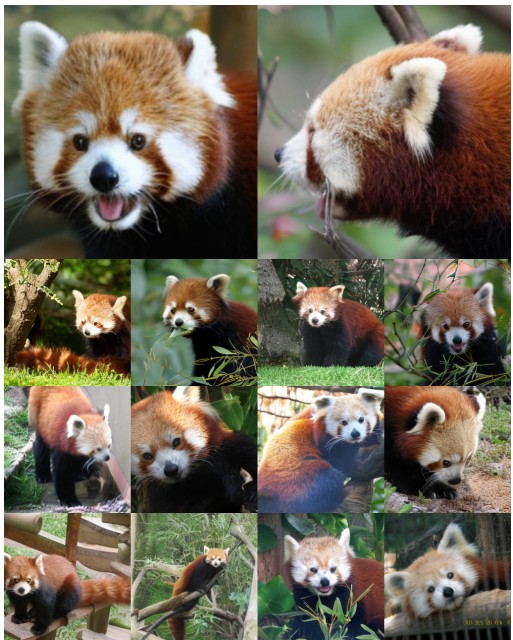

Figure 16: Generated samples of SiT-XL/2+MG on the class lesser panda (387).

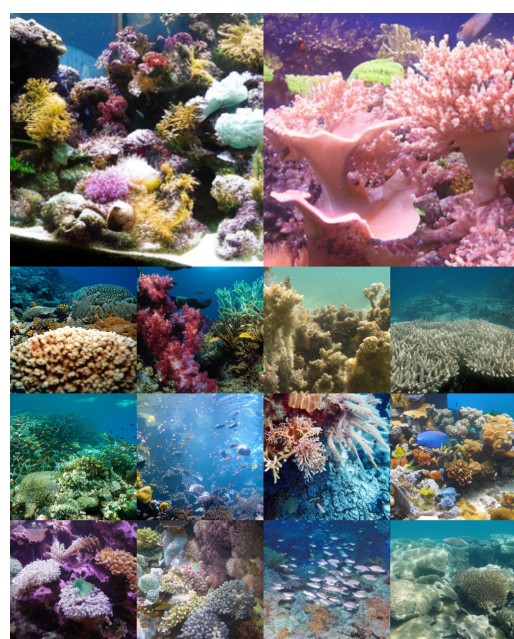

Figure 17: Generated samples of SiT-XL/2+MG on the class coral reef (973).

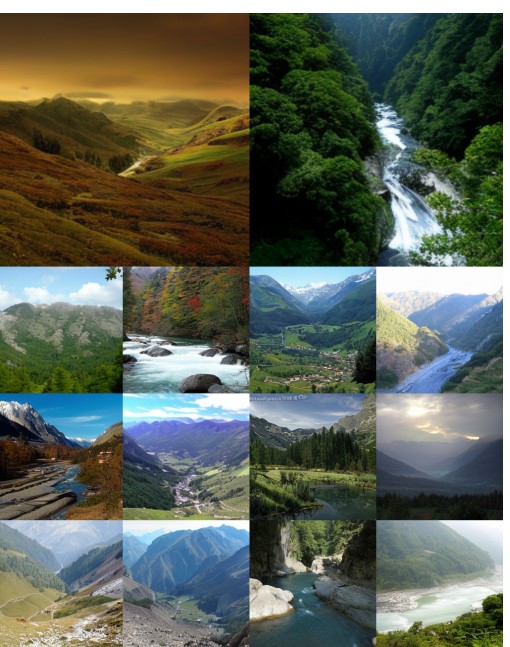

Figure 18: Generated samples of SiT-XL/2+MG on the class valley (979).

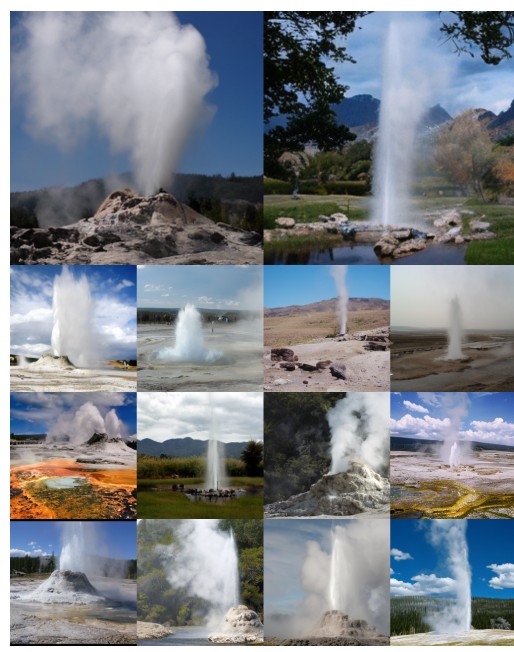

Figure 19: Generated samples of SiT-XL/2+MG on the class geyser (974).

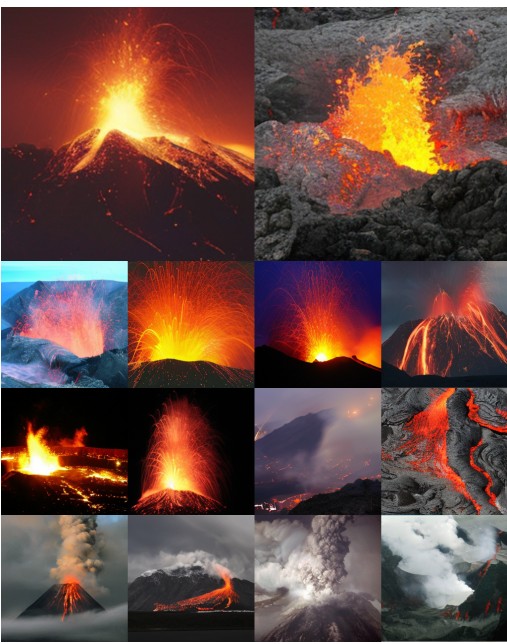

Figure 20: Generated samples of SiT-XL/2+MG on the class volcano (980).

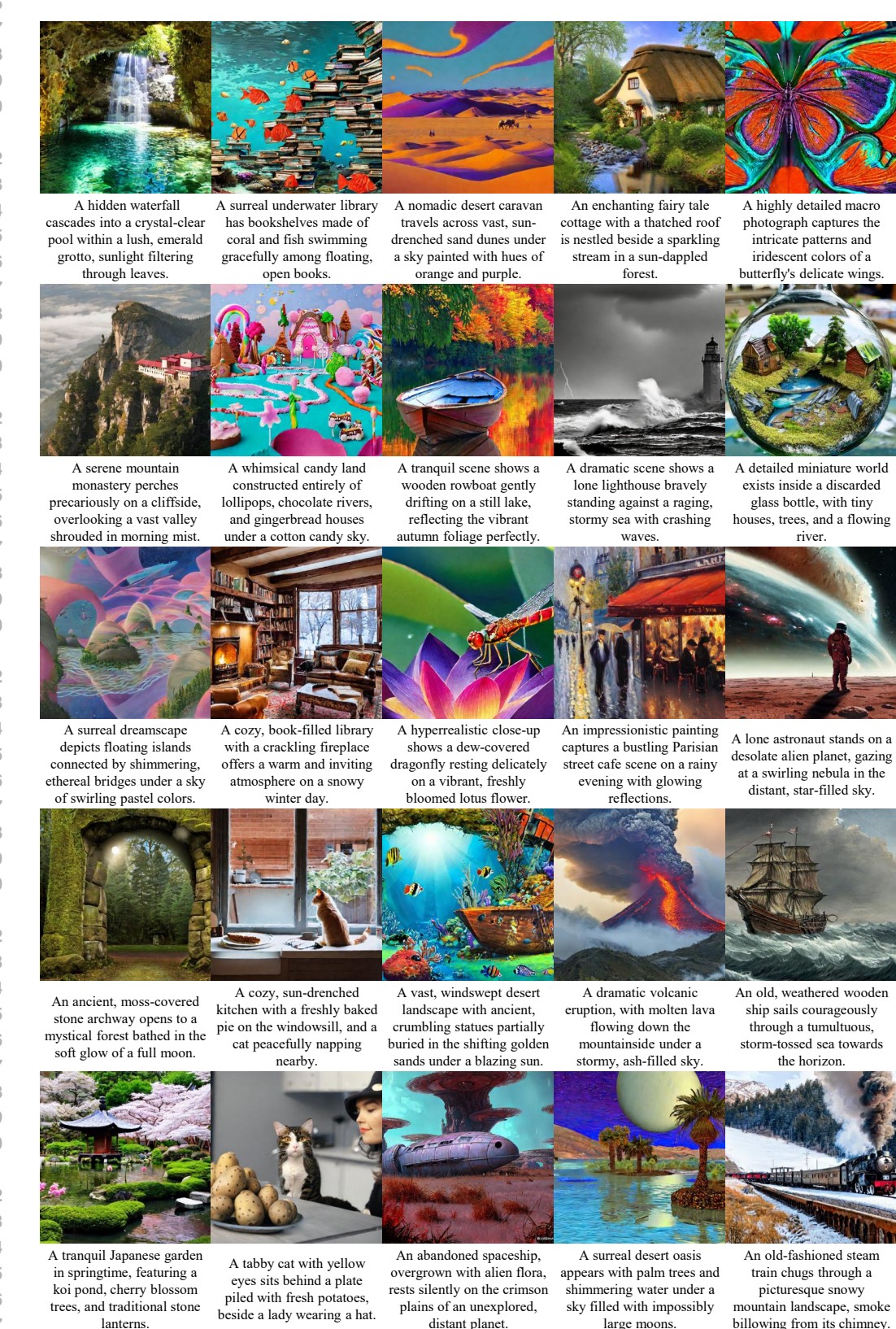

Figure 21: Text-to-image samples of SD1.5+MG.

