# OpenReview forum: "Diffusion Models without Classifier-free Guidance"
_ICLR.cc/2026/Conference — ICLR 2026 Conference Withdrawn Submission_

### Official Review · Reviewer_PGbQ · 2025-10-23

**Soundness:** 2
**Presentation:** 2
**Contribution:** 3
**Rating:** 4
**Confidence:** 4

**Summary:**

The paper proposes a novel training objective that in contrast to CFG avoids learning conditional and unconditional scores separately, and directly learns/estimates the joint score. This leads to compute efficiency while offering similar or improved performance (mostly in terms of generation FID score) when compared to the state of the art recent methods.

**Strengths:**

- The paper is well structured and has reasonably coherent narrative.
- The experiment section is elaborate, and cover different settings and evaluations. Results are promising.

**Weaknesses:**

- Paper definitely benefits from another round of thorough proof-read; there are typos and grammatical error across the document. This has to be improved during the rebuttal.
- I think the justification behind the mathematical derivations in Section 3.1 might not be well established. More specifically, where does Eq (7) comes from? That's not classifier guidance (as posterior should be modeled as $p_\phi$ not $\theta$) and the first term on the right should not be $p(x_t|c)$ but $p(x_t)$. At the same time, I cannot directly relate it CFG. Eq (9) then follows Eq (7) and builds up from there ... this should be discussed and clarified.
- The experimental section also raises some question marks, please see my remarks around that in the next section (questions).

**Questions:**

- Why is the image Fig 3 (b) cropped at the bottom? Is that by design?
- What do you mean by "system-level comparison"?
- In Tables 1 and 2, are you reporting numbers from other references, and that's why there are a number of "-" signs?
- In Tables 1 and 2, the improvement is only computed for the same method not compared to other methods? The claim that the proposed method outperform all other competing baselines is not accurate.
- In Tables 1 and 2, why is MG applied only to DiT and SiT? Can this be applied to other baselines ... if so I do advise updating the Table to demonstrated the impact beyond only these two architectures.
- Table 5 is a bit hard to grasp. Are you trying to compare SD 1.5 with the likes of DALL.E 2? In the second column MG could not be applied?

---

### Official Review · Reviewer_wMfs · 2025-10-30

**Soundness:** 2
**Presentation:** 3
**Contribution:** 2
**Rating:** 2
**Confidence:** 4

**Summary:**

This paper proposes "Model-guidance" (MG), a novel training method for conditional diffusion models that aims to replace the widely used Classifier-Free Guidance (CFG). The core idea is to internalize the guidance mechanism of CFG directly into the training objective. Instead of learning separate conditional and unconditional scores and combining them at inference time (requiring two forward passes), MG trains the model to directly predict a "guided" score function derived from a joint distribution. The authors demonstrate significant performance improvements on ImageNet benchmarks and text-to-image tasks, while claiming to double inference speed by reducing the number of forward passes per step from two to one.

**Strengths:**

The paper addresses a highly relevant and practical problem in diffusion models: the computational cost and potential distributional issues of CFG. The experimental results are comprehensive, showing strong performance gains, particularly on ImageNet, where the method achieves state-of-the-art FID scores. The practical benefit of a 2x speedup in inference is a major potential advantage. Furthermore, the authors provide extensive ablation studies and the method is presented as being simple to implement within existing frameworks.

**Weaknesses:**

Despite the promising results and appealing concept, the paper has critical weaknesses that undermine its contribution and practicality.

Lack of Theoretical Depth and Justification: The theoretical foundation for MG is underdeveloped. The paper presents the loss function as a given, without a rigorous analysis of its convergence properties or the precise distribution it models. The mechanism is presented as a heuristic, and the "self-improvement cycle" lacks a solid theoretical explanation. A method proposing a fundamental shift in the training objective requires a more profound theoretical grounding to be convincing.

Critical Sensitivity and Non-Transferability of Hyperparameter w: The guidance scale w is the most crucial hyperparameter in MG, yet its selection appears highly fragile and dataset- or architecture-specific. As shown in the  Table 6, the performance (FID) varies dramatically with different values of w. This makes the method impractical for real-world applications where extensive hyperparameter sweeps are prohibitively expensive. The proposed auto-tuning scheme is a partial remedy but is computationally costly and itself introduces new hyperparameters (beta_1, beta_2). There is no evidence or discussion of w's transferability across different model scales or datasets, casting serious doubt on the method's generalizability and robustness. A method whose success hinges on such a brittle and hard-to-tune parameter has limited utility.

In conclusion, while the idea is novel and the empirical results on specific benchmarks are strong, the combination of a shallow theoretical analysis and a critical, unresolved sensitivity to hyperparameter w significantly limits the paper's impact and practical value. For these reasons, I cannot recommend acceptance.

**Questions:**

refer to weakness

---

### Official Review · Reviewer_dmaj · 2025-10-30

**Soundness:** 3
**Presentation:** 3
**Contribution:** 2
**Rating:** 4
**Confidence:** 3

**Summary:**

This paper proposes a method to reduce inference costs in the classifier-free guidance (CFG) framework by halving the number of neural network forward passes. The key idea is to train or fine-tune a new neural network that directly “caches” the flow trajectory produced by CFG. During inference, this new network replaces the original extrapolation-based version, effectively saving one forward pass. Empirical results show that the proposed training pipeline accelerates inference without causing a noticeable drop in sample quality.

**Strengths:**

1. The paper is generally well-written and easy to follow.
2. Reducing inference time in diffusion sampling is an important problem, and the proposed method offers a practical approach to halving inference time when classifier-free guidance (CFG) is applied.
3. Although the method is relatively straightforward, the empirical results demonstrate its effectiveness.

**Weaknesses:**

1. While the proposed methods show some promising results, their novelty appears limited. In essence, the method introduces an additional network to record the drift produced by CFG, thereby saving one forward pass during inference.
2. In the introduction, the authors mention several drawbacks of CFG, including the simultaneous modeling of both unconditional and conditional tasks during inference. However, it is unclear how the proposed approach addresses these issues. From my understanding, in addition to the conditioned and unconditioned flows, the proposed network may also need to handle the guidance strength parameter, which could complicate the modeling process further.
3. Regarding the scale-aware networks, the training procedure is not clearly described. The paper mentions that the hyperparameter w is automatically adjusted as training progresses. Does this imply that, once training is complete, the network is well-trained only for a specific value of w? If so, it is unclear how the method enables a trade-off between generation quality and sampling diversity.
4. It is also unclear why the training target in Equation (16) should work as intended. Although the authors present strong empirical results, it would be helpful to provide theoretical insights or analysis to clarify what ground-truth flow this training target induces.

**Questions:**

Please refer to the weakness box.

---

### Official Review · Reviewer_US3H · 2025-10-31

**Soundness:** 3
**Presentation:** 3
**Contribution:** 2
**Rating:** 4
**Confidence:** 3

**Summary:**

This paper proposes Model-Guidance (MG), a novel training objective for diffusion models that eliminates the need for Classifier-Free Guidance (CFG). Unlike CFG, which separately trains conditional and unconditional models and requires two forward passes during inference, MG directly incorporates the posterior probability of conditions into training and turns the diffusion model itself into an implicit classifier. MG achieves comparable or superior generation quality to CFG-based methods while doubling inference speed and simplifying implementation (requiring only minimal code changes). Experiments on ImageNet (256×256 and 512×512) show that MG improves FID scores.

**Strengths:**

1. This paper introduces Model-Guidance (MG), a novel training objective that replaces the traditional Classifier-Free Guidance (CFG) by directly modeling conditional posteriors within diffusion models, thereby unifying conditional and unconditional learning.
2. The proposed method eliminates the need for dual forward passes during inference, doubling generation efficiency while maintaining or improving sample quality.
3. The paper provides a theoretical derivation of MG based on Bayes’ rule and validates it across multiple architectures, including DiT, SiT, and Stable Diffusion, demonstrating consistent performance gains.

**Weaknesses:**

1. From a theoretical perspective, the derivation of the target term in Equation (14) lacks rigor. It is unclear why the first term uses the ground-truth noise ($\epsilon$) instead of the model prediction ($\epsilon_\theta$). The paper does not specify under what assumptions this formulation holds, nor whether it conflicts with the original derivation of Classifier-Free Guidance (CFG).

2. Although the authors employ a stop-gradient mechanism to prevent model collapse, there is no theoretical analysis or guarantee of convergence and stability when the model serves as its own online teacher. Moreover, it is unclear whether MG is trained from scratch or fine-tuned after pretraining—if it starts from scratch, the target signal may be ill-defined or uninformative in the early training stages.

3. During training, the model requires an additional computation of the unconditional prediction $\epsilon_\theta(x_t, t, \varnothing)$ to construct the target $\epsilon'$. Therefore, the increase in computational cost and memory usage compared to vanilla diffusion models has not been sufficiently quantified.

4. In the experiments, the CFG guidance scale is typically fixed, while MG’s scaling parameter $\omega$ is tuned or adaptively adjusted. If the hyperparameter search strategies are not aligned, the comparison may be biased in favor of MG. A fairer evaluation would involve performing equivalent hyperparameter optimization or adaptive scaling for the CFG baseline as well.

**Questions:**

1. During training, MG requires an additional unconditional prediction $\epsilon_\theta(x_t, t, \varnothing)$ to construct the target $\epsilon'$. Could the authors quantify the additional computational and memory costs introduced by this step relative to standard diffusion training?
2. In the experiments, MG’s scaling factor $\omega$ is tuned or adaptively adjusted, while CFG uses a fixed scale. Would the conclusions still hold if CFG were tuned with the same level of effort or adaptive scaling?

---

### Note · Authors · 2025-11-24

I have read and agree with the venue's withdrawal policy on behalf of myself and my co-authors.